# Disease Perception Is Correlated with Health-Related Quality of Life in Patients Suffering from Myelodysplastic Syndromes: Results of the Belgian Be-QUALMS Study

**DOI:** 10.3390/cancers15133296

**Published:** 2023-06-22

**Authors:** Bert Heyrman, Stef Meers, Ann De Becker, Kristien Wouters, Achiel Van Hoof, Ann Van De Velde, Carlos Graux, Dominiek Mazure, Dominik Selleslag, Helena Maes, Jan Lemmens, Marielle Beckers, Dimitri Breems, Sélim Sid, Zwi Berneman, Sébastien Anguille

**Affiliations:** 1Ziekenhuis Netwerk Antwerpen, Department of Haematology, 2020 Antwerp, Belgium; 2Algemeen Ziekenhuis KLINA, Department of Haematology, 2930 Brasschaat, Belgium; 3Department of Haematology, Universitair Ziekenhuis Brussel, 1090 Brussels, Belgium; 4Clinical Trial Center (CTC), CRC Antwerp, Universitair Ziekenhuis Antwerpen, 2650 Edegem, Belgium; 5Algemeen Ziekenhuis Damiaan, Department of Haematology, 8400 Ostend, Belgium; 6Department of Haematology, Universitair Ziekenhuis Antwerpen, 2650 Edegem, Belgiumsebastien.anguille@uza.be (S.A.); 7Department of Haematology, Heilig Hart Ziekenhuis, 2500 Lier, Belgium; 8Centre Hospitalier Universitaire UCL Mont-Godinne, Department of Haematology, 5500 Dinant, Belgium; 9Department of Haematology, Universitair Ziekenhuis Gent, 9000 Gent, Belgium; 10Department of Haematology, Algemeen Ziekenhuis Sint-Jan, 8000 Bruges, Belgium; 11Imelda, Department of Haematology, 2820 Bonheiden, Belgium; 12Gasthuiszusters Antwerpen, Department of Haematology, 2610 Wilrijk, Belgium; 13Department of Haematology, Universitair Ziekenhuis Leuven, 3000 Leuven, Belgium; 14Department of Haematology, Cente Hospitalier Régional Verviers East Belgium, 4800 Verviers, Belgium

**Keywords:** myelodysplastic syndromes, health-related quality of life, disease perception, patient reported outcome, QUALMS, B-IPQ, HRQoL, PRO

## Abstract

**Simple Summary:**

Myelodysplastic syndromes are a heterogenous group of bone-marrow cancers that prelude a possible evolution to acute myeloid leukemia. Most treatments aim at reinstalling bone marrow function and delay evolution to acute leukemia. The disease has a negative impact on health-related quality of life and amelioration is an independent treatment goal. In order to capture the impact of current treatments outside clinical trials we aimed to monitor quality of life during the course of treatment. Another key element in our study was to prospectively assess disease perception and the impact on quality of life. Not reaching our inclusion goal, we cannot make any conclusions or assumptions on our first objective. We can, however, demonstrate a clear significant relation between disease perception and health related quality of life in this patient population that may lead to a positive impact for future patients.

**Abstract:**

Patients with myelodysplastic syndromes suffer from an impaired quality of life that is only partially explained by physical symptoms. In an observational study, we aimed to investigate the impact of current MDS treatments and the influence of disease perception on quality of life. Serial measurement of health-related quality of life was performed by ‘the QUALMS’, a validated MDS-specific patient reported outcome tool. Disease perception was evaluated by means of the Brief Illness Perception Questionnaire (B-IPQ). We prospectively collected data on 75 patients that started on a new treatment and could not demonstrate a significant change in QUALMS score or B-IPQ score during treatment. Six out of eight items evaluated in the B-IPQ correlated significantly with QUALMS score. In this small sample, no significant difference in QUALMS score was found between lower vs. higher risk MDS patients or other studied variables, e.g., targeted hemoglobin showed no correlation with QUALMS score. In daily practice attention must be paid to initial formation of disease perception as it correlates independently with health-related quality of life and does not change during treatment (clinicaltrials.gov identifier: NCT04053933).

## 1. Introduction

Myelodysplastic syndromes (MDS) are a heterogenous group of hematological malignancies. They are driven by a plethora of molecular aberrations and come with a variable degree of bone marrow failure and risk of progression to acute myeloid leukemia [1]. The clinical course is dominated by a reduced performance status and impaired health related quality of life (HRQOL). However, presentation with asymptomatic disease is not uncommon [2]. Since the only curative option of allogeneic stem cell transplantation is reserved for a minority of patients, maintenance or improvement of HRQOL is always a treatment goal. In lower risk disease the issue is more prominent since the disease does not affect overall survival to a great extent. Nevertheless, since a cure is not an achievable goal in the majority of high risk MDS patients with a median survival of less than 2 years, HRQOL remains a primary concern [3].

Restrictions in HRQOL are the result from a high symptom burden and can only be partially explained by anemia. Assessment of HRQoL by patient reported outcome (PRO) can provide insight in the patient’s perspective and aid physicians in orchestrating therapeutic efforts. This is not only necessary to ameliorate the daily life of MDS patients, restrictions in HRQoL may also prelude a less favorable clinical outcome [4]. An objective HRQoL assessment can also provide a response criterion to treatment and clears the path to implement HRQOL as a safety aspect as we argued earlier [5]. The growing interest in HRQoL has led to the development and propagation of disease specific PRO-tools that have become standard content in interventional trials with new medications. Outside clinical trials, PRO-tools are seen as time consuming and have a reputation of a low benefit/effort ratio. Therefore impact of current MDS treatments in real life remains a topic of debate [3].

The mental experience of a disease is largely influenced by the disease perception [6]. In other forms of cancer illness perception is found to be an independent prospective predictor of HRQoL [7]. A more negative perception around the time of diagnosis resulted in lower future HRQoL-scores. Apart from the impact on HRQoL, illness perception can also correlate with mortality in cancer survivors [8]. No data can be found on the evolution of disease perception in MDS or the possible influence on HRQoL. The main objective of this prospective observational study was to investigate the impact of current MDS-treatments on HRQOL and collect information on disease perception. A secondary objective was to assess different clinical features e.g., target hemoglobin on HRQOL outcome.

## 2. Materials and Methods

### 2.1. Participants

Newly diagnosed patients with MDS were invited to participate in the study in case they started a treatment. Patients who were not in need for treatment were not retained for invitation. Known patients with MDS who already had a previous therapy were invited in case they started on a new treatment. Previously diagnosed patients who were only on a continuous treatment that started before the study could not participate. The primary objective was to include 350 patients spread over 7 treatment groups, i.e., packed cells transfusion, erythropoiesis-stimulating agents (ESA), deferoxamine, deferasirox, 5′azacytidine, lenalidomide and intensive chemotherapy. After one year, the study was amended. Following the amendment, iron chelators were considered as one treatment group, luspatercept was added as a separate group and study-drugs were allowed.

### 2.2. HRQOL Measurement

Patients filled out an MDS-specific measure of QoL (the Quality of Life in Myelodysplasia Scale, [QUALMS]) at the start of a new treatment and at 4, 12 and 24 weeks into this treatment. The instrument was validated in the US and later in Europe [9,10]. The QUALMS consists of 33 questions with a five-level answer. The final score is outlined on a scale of 0 to 100 (a higher score implies a better HRQOL). Additionally, at the end, 5 single-item questions are presented that are not included in scoring as they do not apply to all patients. The questionnaire comprises three sub scores. The QUALMS-P measures the physical burden of the disease in 14 items. The psychological impact of disease is evaluated in 11 items corresponding to the emotional burden subscale (QUALMS-E). A 3-item benefit-finding subscale (QUALMS-BF) assesses positive life changes as a result of diagnosis [9]. The questionnaire has showed consistency for the total score (Cronbach’s alpha, 0.92), the physical burden subscale (Cronbach’s alpha, 0.91) and the emotional burden subscale (Cronbach’s alpha, 0.91). Only for the benefit-finding subscale consistency was low (Cronbach’s alpha, 0.54) [10].

### 2.3. Illness Perception Measurement

Simultaneously filling out the QUALMS at the before mentioned timepoints, patients filled out the Brief Illness Perception Questionnaire (B-IPQ). The B-IPQ is a 9-item scale developed to rapidly assess the cognitive and emotional representations of any disease, not specifically hematological malignancies [11]. The first 8 items are scored using a 0–10 scale, and the last item uses an open response. A total score can be calculated on a scale of 0 to 80, with inverse calculation of questions 3, 4 and 8, or each question can be studied separately or analyzed in domains (cognitive illness representations and emotional illness representations). The B-IPQ was repeatedly validated in different patient populations as well as in cancer research (Cronbach’s alpha total score, 0.85; Cronbach’s alpha for cognitive illness representations, 0.80; Cronbach’s alpha for emotional illness representations, 0.83) [12].

Both the QUALMS and B-IPQ were filled out on paper and transferred to a digital database by trained personnel. Sampling was performed at different timepoints to minimize copy mistakes.

### 2.4. Demographic and Clinical Parameters

Apart from the date of birth and gender we collected personal information on the housing and relational status, financial worries, whether or not the patient had children with whom they were in contact and if the patient received an information brochure at diagnosis or not. A brief medical history was collected in terms of cardiac failure, respiratory failure, ECOG status and recent hospital admissions. Disease characteristics were collected regarding peripheral blood levels, bone-marrow characteristics, cytogenetic results and molecular results to allow calculation of specific MDS risk scores for each patient. Apart from this, we asked for the goal of the treating physician concerning the target Hb level, transfusion limit for each patient and which MDS specific treatment was started. From the second timepoint (i.e., at 4 weeks) onwards, we collected information on whether or not the patient was considered a responder at the discretion of the treating physician.

### 2.5. Statistical Analysis

Statistical analysis was performed in R version 4.1.3. This study originally aimed to include 350 patients, 50 per treatment group, to be able to show a significant difference of 7.5 points (SD 15 points) in QOL between baseline and the 3 follow-up visits in each of the treatment arms separately. The estimate of the standard deviation was based on the QUALMS validation study [9]. A Bonferroni correction for multiple testing (3 comparisons per treatment arm) was applied and a dropout rate of 10% was accounted for in the final number of 50 patients per arm. As the study was ended prematurely, the statistical analysis plan was reconsidered, and focus was shifted to descriptive statistics and evolution over time in the whole sample, with only minimal inspection of subgroups. Patient characteristics are summarized as numbers and percentages for factors and mean and standard deviation for numeric variables. Normality was assessed by means of Shapiro–Wilk test in combination with visual inspection of QQ plots. Baseline differences in QUALMS and IPQ scores between subgroups were studied with a *t*-test. Evolution of QUALMS and IPQ scores and subscores over time were studied in linear mixed effects models, with fixed effect ‘time’ (4–12–24 weeks) as a categorical variable and a random intercept per patient, accounting for the dependency between observation of the same individual. In case of a significant time-effect, post hoc pairwise comparisons were made between the different visits. Bonferroni–Holm correction was applied to correct for multiple testing. Estimates, standard errors, raw and corrected *p*-values were reported. Subgroup differences (newly diagnosed/new treatment and responder/non-responder) in QUALMS over time were studied by adding the subgrouping as a fixed factor to the linear mixed models. Non-parametric spearman correlation coefficients and corresponding *p*-values were calculated to study the association between QUALMS, IPQ, IPSS and IPSS-R. For the association between change in IPQ and change in QUALMS, correlation coefficients were calculated based on linear mixed effects models. Associations are visualized in scatterplots with locally estimated scatterplot smoothing (loess smoothing).

## 3. Results

The study opened 1 September 2019 and was terminated after 2 years, before reaching the inclusion target.

### 3.1. Patient and Disease Characteristics

We collected data on 75 patients (23 female, 52 male). The majority had an advanced age (median age 79 years (55–94)). Most of the patients were housed at home (93.2%), lived with a partner (72.2%) and did not report financial worries (97.1%). The majority (81%) had children with whom they had contact. Follow-up with a psychologist was a rarity and only reported by 4.3% of patients. A recent infection was reported by 6 patients at week 4, by 6 patients at week 12 and by 5 patients at week 24. Recent hospitalization (>2 days) occurred in 5 patients at week 4, in 6 patients at week 12 and in 4 patients at week 24. Distribution of patients by IPSS-classification revealed a low risk in 37%, an intermediate risk −1 in 32% and −2 in 16% and a high risk in 8%. Scoring by IPSS-R showed a very low risk in 5%, low risk in 37%, intermediate risk in 21% and a high/very high risk in 19% and 11%, respectively. Risk-scores could not be calculated in 7%. In our examined sample 49 patients had a new diagnosis of MDS while 26 received an MDS-related therapy preceding study participation. Most represented were patients receiving ESA (*n* = 39). Other represented treatments were azacytidine (*n* = 20), transfusion without any other treatment (*n* = 6), lenalidomide (*n* = 3), luspatercept (*n* = 2), iron chelation (*n* = 2), intensive chemotherapy (*n* = 1), venetoclax (*n* = 1), investigational drug (*n* = 1). Demographic characteristics of the patients could not be further examined towards their relation with QUALMS score or B-IPQ due to the small sample size and homogeneity of the sample. Experiencing a recent infection or hospitalization was not further explored since the high level of infections during COVID pandemic could be related to missing data. No significant difference in HRQoL could be demonstrated between lower and higher risk MDS based on IPSS or IPSS-R and QUALMS score (mean low/int-1 IPSS 65.9 (SD 12.9) vs. int-2/high 60.0 (SD 13.2), *p* = 0.11 and mean very low/low/intermediate IPSS-R: 64.9 (SD 12.2) vs. high/very high 63.3 (SD 15.3), *p* = 0.65, respectively). A Spearman rank correlation between baseline Hb and QUALMS score did not show any significant relation (rho = 0.05, *p* = 0.72).

### 3.2. Patients with MDS Suffer from Impairment in HRQoL That Remains Stable during Treatment

In the patient group studied, 37% filled out both questionnaires at all prescheduled timepoints. Intermittent missing questionnaires were noted in 17% and dropouts were seen at week 4 (11%), week 12 (20%) and week 24 (15%). Overall, we registered a mean QUALMS score of 64.7 (SD 12.9) at baseline. The score did not vary significantly over time (*p* = 0.31) and thus no change in HRQOL was noted during treatment (Figure 1). The same observation was made for the QUALMS-E (*p* = 0.52) and QUALMS-P (*p* = 0.13). The change in QUALMS-BF over time was borderline non-significant (*p* = 0.059). In a pairwise analysis a significant increase was noted at week 24 vs. baseline (estimated difference in mixed model: +7.9 (se 3.4) *p* = 0.022) and vs. week 4 (+7.6 (se 3.4) *p* = 0.028). After correction for multiple testing (Bonferroni–Holm), this significance was not retained (*p* = 0.13 and 0.14, respectively).

### 3.3. Disease Perception Does Not Change during the Course of Treatment

We prospectively analyzed the B-IPQ response per question. The overall impact of the disease is assessed in the first item and showed a median severe (baseline score 5.8) impact of the disease with no significant change during treatment. The timeline or interpretation of the duration of the disease is assessed in the second item. Most patients felt the disease was likely to continue for a long period (baseline score 7.8). Control over the disease resulted in a median score of 4.5 at baseline.

In item 4, looking at the perception of usefulness of current therapy, there was a significant increase between baseline (score 7.5) and week 12 (score 8.0) (estimated difference in mixed model: +0.67 (SE 0.30) *p* = 0.025) that was not seen at week 4 (score 7.8) (estimated difference +0.18 (se 0.27) *p* = 0.52) and did not hold afterwards (baseline vs. week 24 difference +0.07 (SE 0.32) *p* = 0.82). In item 5 (experience of symptoms), we found a significant decrease over time from baseline (score 5.0) to week 4 (score 4.1) (estimated difference −0.91 (SE 0.40) *p* = 0.025) that remained stable but did not change from week 4 onwards. This reflects a decline in physical symptoms early at the start of treatment but no further gain following week 4. Patients are concerned about the disease (item 6, baseline score 7.1) and feel they understand the disease more or less (item 7, baseline score 6.4). The emotional impact of disease was meaningful (baseline score 5.4). All results can be found in Table 1.

Possible causes of diseases are assessed in an open response item, asking patients what they think are the three most important causal factors of their disease. Answers could be categorized into one of four groups, i.e., external factors (*n* = 22), genetic predisposition (*n* = 7), age (*n* = 21) and unknown cause (*n* = 16). External factors varied widely (e.g., air pollution, previous chemotherapy, emotional stress, nutrition, smoking, contact with chemicals, physical activity, etc.). We could not demonstrate any significant difference in QUALMS score between the four groups.

### 3.4. Disease Perception Is Correlated with Health-Related Quality of Life

Correlation between QUALMS score and B-IPQ score per item was investigated at baseline. We found a significant correlation between QUALMS score and most B-IPQ items (Figure 2).

The QUALMS-emotional and QUALMS-burden correlated significantly with the same B-IPQ items as the general QUALMS score. The strongest correlation was found between the concern about the disease (B-IPQ item 6) and QUALMS score and the emotional impact of the illness (B-IPQ item 8) and QUALMS score (respective rho = −0.44 and −0.48, both *p* < 0.001). The overall impact of the disease (B-IPQ item 1), the perception of disease control (B-IPQ item 3) and the experience of symptoms (B-IPQ item 5) correlated with QUALMS score (*p* < 0.01). The confidence in treatment (B-IPQ item 4) associated significantly with QUALMS score (*p* = 0.082) but the estimation of duration of disease (B-IPQ item 2) and the impression of understanding the disease (B-IPQ item 7) were not (*p* = 0.31 and *p* = 0.77 respectively). Correlation coefficients and corresponding *p*-values for the relation between B-IPQ and QUALMS score and subscores at baseline can be found in Table 2.

A significant amelioration in the experience of symptoms (B-IPQ item 5) between the start of treatment and week 4 into treatment without an overall change in QUALMS score was detected for the total sample. Further investigation, in which change in QUALMS score was linked to change in B-IPQ score per patient, showed a significant correlation (r = 0.31, *p* < 0.001) between improvement of symptoms (decreasing B-IPQ score in item 5 compared to baseline) and increase in QUALMS score (Figure 3).

### 3.5. No Significant Relation Was Found between HRQoL and Explored Interventions or Treatment Status

In the group of patients with a new diagnosis, 25 patients received an information brochure on the disease and 18 did not, and 5 patients did not respond to this question. No correlation could be demonstrated with B-IPQ item 7, ‘How well do you feel you understand your disease?’ (mean 6.28 (SD 2.57) vs. 6.11 (SD 2.19) *p* = 0.82).

In the discretion of the treating physicians, 75.8% were considered responders to ESA. QUALMS scoring showed no substantial difference between responders and non-responders to ESA (*p* = 0.68). The small group had not enough power for a firm conclusion. In an analysis of the total sample comparing responders to any treatment vs. non-responders, no significant difference in QUALMS score could be demonstrated (*p* = 0.37).

We asked the hematologist the specific Hb target for each patient. This was a Hb level of 8 g/dL in 11.4%, 9 g/dL in 30%, 10 g/dL in 42% and 11 g/dL in 15.7%. No significant correlation could be found between target Hb level and QUALMS score (Spearman rho = 0.19, *p* = 0.13).

We also compared the QUALMS score of patients that had a new diagnosis versus patients that were diagnosed in the past but started on a new treatment. The QUALMS score appears higher in the group with a previous diagnosis (mean 69.8 (SD 14.7) versus 63.7 (SD 17.1)), but this difference was not significant (*p* = 0.32), which could be due to the small sample size.

## 4. Discussion

This prospective observational trial aimed to investigate the impact on quality of life of initiating any therapy for MDS and study the correlation with disease perception of MDS patients. A weakness of our study is the low inclusion rate. The timing of the study (opening September 2019) led to insufficient accrual, similar to what was observed in cancer control and prevention trials during the COVID-19 pandemic [13]. As a consequence, for not meeting the foreseen inclusion target the funding of the study was withdrawn and closed prematurely. The same argument came back in terms of missing data that were a consequence of diminished hospital visits and permitted time in the hospital and waiting room where most of the questionnaires were filled out. In total, we collected data on 75 patients. We used the QUALMS, an MDS-specific PRO-tool, to investigate prospective change in HRQoL and seek correlation with disease perception. This paper reports the first results of the use of the QUALMS in Belgian MDS patients.

Looking at the complete group of patients, an impaired MDS-related QoL was observed, similar to the QUALMS registration study (mean QUALMS score 64.7 (SD 12.9) vs. mean QUALMS score 67.2 (SD 15.2) in the registration study) [9]. Unfortunately, due to low inclusion numbers, we were unable to confirm or reject the hypothesis that lower risk MDS subgroups have significantly higher QUALMS scores vs. higher risk MDS patients [9]. Correlation between MDS risk group and HRQoL remains debatable since other studies could not demonstrate correlation between MDS risk groups and HRQoL [14].

The impairment in HRQOL detected at the start of one of the treatments did not deteriorate during treatment. With current therapies, HRQoL remained stable, which is seen as a primary goal of MDS treatments. The small sample size per treatment group does not permit conclusions per treatment group, which was a primary objective of this study. An effort was made for the largest group of patients receiving ESA. Compared to historic data, a higher percentage of responders was noted. This analysis was underpowered and resulted in a non-significant difference between the two groups. In a further analysis of the total sample, responders to treatment versus non-responders, we could not demonstrate a difference in QUALMS score. The variety in type of treatment and small scale of the sample does not permit any conclusions on this topic.

Transfusion protocols differ among institutions and treating physicians. A higher hemoglobin threshold might contribute to improvement in HRQoL, as was shown in a small patient group (*n* = 38) [15]. A more liberal transfusion threshold comes with more frequent transfusions. Transfusion dependency and secondary iron overload are correlated to a reduced overall survival (OS) [16]. In our patient group, we found no significant relation in Hb level and QUALMS score or Disease perception and Hb level at baseline. We could not demonstrate a significant correlation between QUALMS score and target Hb level (rho = 0.19). This finding adds to the need of larger studies that include different parameters (i.e., HRQoL, OS, cost for society, etc.) to answer the question of the optimal transfusion policy.

In our opinion, analysis of the B-IPQ responses per question holds a more a meaningful approach instead of the total B-IPQ score. The table of illness perception scores (Table 1) allows comparison to the scores outlined in an earlier systematic review by Broadbent et al. [6]. Compared to previous studies using the B-IPQ, our results are in line with other cancer-related trials. In our patient group, disease perception is installed early in the disease course and apart from physical symptoms, perception does not change significantly during treatment.

We observed a significant relation between most B-IPQ items (1, 3, 4, 5, 6 and 8) and overall QUALMS score and subscores QUALMS-emotional and QUALMS-burden. QUALMS scoring is an in-depth assessment of item 1 in the B-IPQ, assessing how much the disease affects the patient’s life. A strong correlation is therefore expected. A lower level of control over their disease (B-IPQ item 3) was felt by the majority of patients and correlated significantly with lower QUALMS scores. A sensation that might be altered by giving patients more responsibility during their treatment follow-up. Confidence in the help of the current treatment was high (B-IPQ item 4) and more confidence contributed significantly to higher QUALMS scores. Disease symptoms were mostly perceived as non-severe with a mean score of 5 at baseline and significant decrease to 4.1 at week 4. This points to the debilitating nature of disease symptoms that are not perceived as acute and life-threatening but correlate significantly with HRQoL. A correlation analysis over all timepoints showed a significant association between improvement of physical symptoms and gain in HRQoL. However, the significant decrease in symptom score was not translated to an overall increase in QUALMS.

Previous studies have noted that emotional issues are often perceived as more invalidating than physical symptoms [17]. Although most patients suffered from a lower risk disease that holds a better prognosis and has less impact on life expectancy, concern about the disease was high (B-IPQ item 6). A higher level of concern correlated significantly with lower QUALMS scoring. Apart from installing trust in the foreseen treatment, hematologists must pay attention at installing trust in the expected natural behavior of the disease, thereby diminishing this concern. With a mean score of 5.4, the emotional impact of the diagnosis (B-IPQ item 8) cannot be underestimated,. Once treatment was started, no change was noted during treatment and correlation with QUALMS score was significant.

Most patients were realistic in their expectations that the disease would continue life-long (B-IPQ item 2, mean score 7.8). In the sub-analysis, this item had no significant correlation with QUALMS score, indicating that patients with different disease expectations did not necessarily experience a better HRQoL. The majority of our patients were in the grey zone in terms of understanding their disease (B-IPQ item 7). This item showed no correlation with QUALMS score. Receiving an information brochure at diagnosis did not result in a better disease understanding since we did not find any significant difference in B-IPQ item 7 between patients who received a brochure at diagnosis vs. non-brochure receivers.

## 5. Conclusions

We conclude that early at the start of treatment, the sensation of disease control, confidence in treatment, physical symptom burden, concern about the disease and emotional impact of the disease are installed and correlate significantly with quality of life in MDS patients. In future communication to MDS patients, attention must be paid to the emotional impact of the diagnosis. This forces us to choose our initial wording carefully. In a later stage, installation of control and confidence in treatment and in the natural behavior of the disease is important. This might alleviate the patient’s concern about the disease and result in an ameliorated HRQoL.

## Figures and Tables

**Figure 1 cancers-15-03296-f001:**
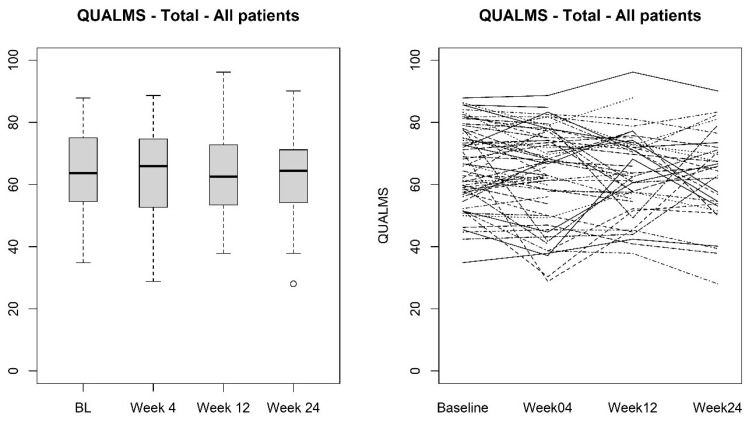
QUALMS total score evolution over time.

**Figure 2 cancers-15-03296-f002:**
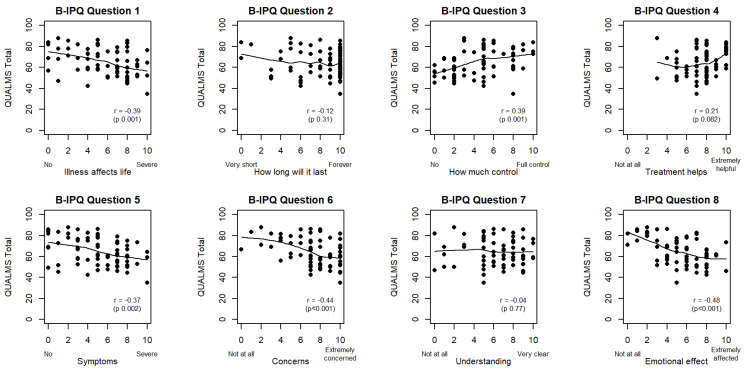
Qualms total score correlation with B-IPQ per item: scatterplots with loess smoothing and Spearman rank correlation coefficients. Significant correlations were found between QUALMS and B-IPQ items 1, 3, 4, 5, 6 and 8.

**Figure 3 cancers-15-03296-f003:**
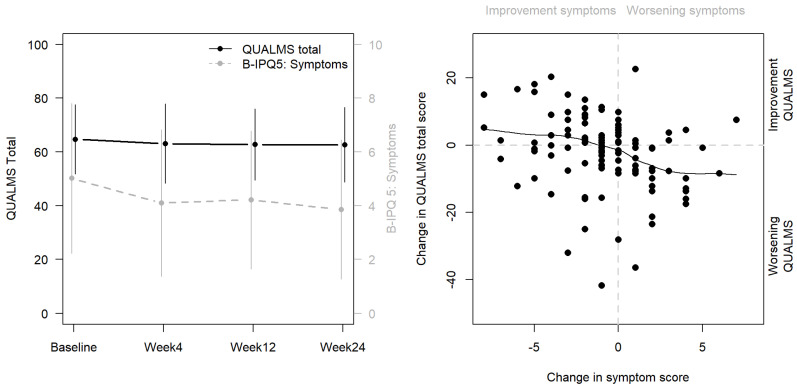
Left panel: evolution in QUALMS score and B-IPQ item 5 for the total sample (mean ± 1 standard deviation). Right panel: scatterplot of the change in QUALMS score versus the change in B-IPQ item 5 (compared to baseline). Positive differences indicate an increase in score. Loess smoothing and scatterplot show a correlation between symptom improvement and increase in QUALMS scoring.

**Table 1 cancers-15-03296-t001:** B-IPQ scoring per question at each timepoint.

B-IPQ Item		Baseline (*n* = 67)	Week 4 (*n* = 59)	Week 12 (*n* = 45)	Week 24 (*n* = 35)
How much does your illness affect your life? (“0” no effects at all—severely affects my life “10”)	Mean (sd)	5.8 (2.8)	5.7 (2.9)	5.9 (2.8)	6.1 (3.1)
How long do you think your illness will continue? (“0” a very short time—forever “10”)	Mean (sd)	7.8 (2.7)	8.0 (2.3)	7.4 (3.3)	8.6 (2.4)
How much control do you feel you have over your illness? (“0” absolutely no control—extreme amount of control “10”)	Mean (sd)	4.5 (2.9)	4.8 (2.8)	5.4 (2.8)	4.5 (3.1)
How much do you think your treatment can help your illness? (“0” not at all—extremely helpful “10”)	Mean (sd)	7.5 (1.7)	7.8 (1.6)	8.0 (1.5)	7.5 (2.8)
How much do you experience symptoms from your illness? (“0” no symptoms at all—many severe symptoms “10”)	Mean (sd)	5.0 (2.8)	4.1 (2.8)	4.2 (2.7)	3.9 (2.6)
How concerned are you about your illness? (“0” not at all concerned—extremely concerned “10”)	Mean (sd)	7.1 (2.4)	6.6 (2.8)	6.7 (2.6)	6.9 (2.9)
How well do you feel you understand your illness? (“0” don’t understand at all—understand very clearly “10”)	Mean (sd)	6.4 (2.5)	6.6 (2.9)	6.2 (2.6)	5.7 (3.2)
How much does your illness affect you emotionally (“0” not at all affected emotionally—extremely affected emotionally “10”)	Mean (sd)	5.4 (2.4)	5.3 (3.1)	5.4 (3.1)	5.3 (3.0)

**Table 2 cancers-15-03296-t002:** B-IPQ correlation with QUALMS at baseline.

	QUALMS Final		QUALMS Emotional		QUALMS Benefit		QUALMS Burden	
B-IPQ1 (life-affecting)	r = −0.39 (*p* = 0.001)	**	r = −0.27 (*p* = 0.026)	*	r = 0.28 (*p* = 0.022)	*	r = −0.48 (*p* < 0.001)	***
B-IPQ2 (duration)	r = −0.12 (*p* = 0.31)		r = −0.08 (*p* = 0.50)		r = 0.29 (*p* = 0.018)	*	r = −0.18 (*p* = 0.14)	
B-IPQ3 (control)	r = 0.39 (*p* = 0.001)	**	r = 0.36 (*p* = 0.003)	**	r = −0.04 (*p* = 0.74)		r = 0.35 (*p* = 0.004)	**
B-IPQ4 (treatment)	r = 0.21 (*p* = 0.082)	˙	r = 0.16 (*p* = 0.202)		r = 0.16 (*p* = 0.19)		r = 0.14 (*p* = 0.25)	
B-IPQ5 (symptoms)	r = −0.37 (*p* = 0.002)	**	r = −0.27 (*p* = 0.028)	*	r = 0.14 (*p* = 0.25)		r = −0.43 (*p* < 0.001)	***
B-IPQ6 (concerns)	r = −0.44 (*p* < 0.001)	***	r = −0.50 (*p* < 0.001)	***	r = 0.17 (*p* = 0.17)		r = −0.30 (*p* = 0.014)	*
B-IPQ7 (understanding)	r = −0.04 (*p* = 0.77)		r = −0.15 (*p* = 0.21)		r = 0.38 (*p* = 0.002)	**	r = −0.07 (*p* = 0.55)	
B-IPQ8 (emotional)	r = −0.48 (*p* < 0.001)	***	r = −0.39 (*p* = 0.001)	**	r = 0.14 (*p* = 0.27)		r = −0.42 (*p* < 0.001)	***

*** *p* < 0.001, ** *p* < 0.01, * *p* < 0.1.

## Data Availability

The dataset for this study can be obtained from the corresponding author upon reasonable request.

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
