# Peer review of "Disease Perception Is Correlated with Health-Related Quality of Life in Patients Suffering from Myelodysplastic Syndromes: Results of the Belgian Be-QUALMS Study"

_cancers, 2023, doi:10.3390/cancers15133296_

Round 1
Reviewer 1 Report
This is a paper describing the work of Heyrman et al. about quality of life (QoL) in Myelodysplastic syndromes (MDS) patients, an important topic that is gaining more and more attention. The researchers set out to evaluate the quality of life in patients with MDS. Initially it was to be a large study of 350 patients divided into 7 groups of 50, each group given a different treatment modality. In the end, however, owing to the Covid-19 epidemic and funding complications, the study was terminated early and only 75 patients were enrolled.
The power analysis determined that to see a difference in QoL, 50 patients would be needed in each group. This makes it much more difficult to interpret the results with only 75 patients. For example comparing the higher risk to the lower risk MDS.
Even fewer patients completed all of the questionnaires at all the time points.
For this reason, negative results are very hard to interpret.
What is significant, as stated in the title of the manuscript is the relationship between QoL and the perception of disease as measured by the brief illness perception questionnaire (B-IPQ). This is important, because the health care provider may have an impact here. However, a very small portion of the results is devoted to this. Here is where the authors should tell the story. Perhaps graphs to demonstrate the relationship and the correlation for the most important results would be useful.
Simply giving the number of B-IPQ that demonstrated a correlation with QoL is confusing. There should be a short description for at least some of them. For example, patients felt that their illness would continue forever (B-IPQ number 2, scores around 7.5-8), and this had no impact on QoL. Yet, few patients felt that they had control over their illness (B-IPQ number 3, scores around 4.5-5) and this did have a significant impact on their QoL.
There should also be a short description of what the QUALMS scores – what difference is between final, emotional, benefit, and burden. Most readers understand QoL, but may not appreciate the specifics of QUALMS.
Minor points:
1) Tables 1 and 2, separation is confusing. I would put horizontal lines to separate.
2) Figure – no caption is presented.
Much of the text is not written in clear enough English. I would suggest improvement.
Author Response
We would like to thank the reviewer for the contributing feedback.
We do agree negative results are very hard to interpret and therefore should not be the focus of this manuscript. In this perspective your comment on extending the results on what was significant i.e. the relation between B-IPQ and QUALMS was taken to heart. More in depth analysis were performed, and specifically on the relation between change in physical symptoms and change in QUALMS resulted in a modified conclusion. Two extra figures were implemented to visually support the results.
To ease reading and avoid the necessity of going back and forward on the numbering of the B-IPQ items, the focus has been brought on the content of the items as suggested.
QUALMS subscores were clarified as suggested.
Minor points:
1) Tables 1 and 2, separation is confusing. I would put horizontal lines to separate. – This was adapted accordingly.
2) Figure – no caption is presented. – All figures have corresponding captions.
Reviewer 2 Report
The purpose of this prospective observational study, conducted on patients with Myelodysplastic Syndromes (MDS), evaluated before and after starting a new treatment, was to analyze the impact of the treatment and the influence of disease perception on Health- Related Quality of Life (HRQoL). The HRQoL was evaluated using the QUALMS , a validated MDS-specific patient reported outcome tool., while the disease perception was evaluated by means of the Brief Illness Perception Questionnaire (B-IPQ).
I have already performed a review of the same manuscript for another journal a short time ago: the Authors accepted my suggestions by making some changes recommended by me. At this point I had considered the manuscript acceptable for publication, albeit with the limitations (recognized by the authors in the discussion) due to the small sample size. As the Authors report, the primary objective of this study was to include 350 patients, spread over 7 treatment groups. Unfortunately, as the study was conducted during the period of the COVID pandemic, the 'accrual was insufficient, and only 75 patients were enrolled, and only 37% of them filled out both questionnaires at all prescheduled time points. As a consequence the study was prematurely closed, as the funding of the study was withdrawn.
Despite the modifications made by the Authors, the manuscript was not accepted by the previous journal, but in my opinion the manuscript thus modified can be accepted for publication.
Author Response
Thank you for previous revision and positive contribution to this manuscript.
Reviewer 3 Report
Interesting article about an important topic. Patient reported outcomes and quality of life measures are increasing incorporated as end points in trials and clinical practice. The authors showed that some correlation between patient's quality of life and how they perceive their disease. They showed important points that patients form early disease perceptions that tend to hold through time which emphasizes the importance of good patient-physician interactions especially closer to diagnosis. Patients have little control over their disease which is why I agree it is important to involve them in decision making.
however I am concerned about their conclusions which perhaps in a bigger sample of patients, will be look different.
1- small sample size which in quality of life observational studies can be lead to poor results which lowers the scientific impact of the paper. Many patient dropped eventually. The authors could not control for important factors that can confound the results for example: infection rate, hospitalization. The sample of patients was very homogenous from a socioeconomic standpoint which has major impact on quality of life.
2- I am not sure that patient with MDS can be grouped into the seven described groups as patients can be treated with combination regimens.
When you mention after 1 year other treatments were allowed, what exactly do you mean?
3-When you describe patient characteristics. Howe many had frequent hospitalizations or frequent infections?
4- were some patients with higher risk MDS evaluated for transplant? How did this affect their disease perception?
5- In the discussion, instead of referring to the questionnaire items with numbers try to explain what this means so that readers don't have to go back and forth.
Overall, I think the study would have resulted in more powerful outcomes with a larger sample size and different statistical methods.
The article needs some sentences rephrased. Well written overall.
Author Response
We would like to thank the reviewer for the contributing feedback.
- small sample size which in quality of life observational studies can be lead to poor results which lowers the scientific impact of the paper. Many patient dropped eventually. The authors could not control for important factors that can confound the results for example: infection rate, hospitalization. The sample of patients was very homogenous from a socioeconomic standpoint which has major impact on quality of life.
The small sample size is indeed a strong limitation of this study, making negative results very hard to interpret. We originally planned to investigate on confounding factors such as infection rate and hospitalization and included as much as possible in our CRF. The small sample size however did not permit any further analysis of this confounding factors. We therefore extended the results on what was significant i.e. the relation between B-IPQ and QUALMS. More in depth analysis were performed, and specifically on the relation between change in physical symptoms and change in QUALMS resulted in a modified conclusion. Two extra figures were implemented to visually support the results. Socioeconomoic status has indeed an impact on HRQoL. In this perspective we can only compare our results with the results presented in the original registration study.
- I am not sure that patient with MDS can be grouped into the seven described groups as patients can be treated with combination regimens.
Outside clinical trials combination regimens are an exception and not reimbursed in Belgium. Since our trial aimed at real-life data, treatment groups where chosen based on reimbursed treatments at the time of conceptualization.
When you mention after 1 year other treatments were allowed, what exactly do you mean?
Patients treated with study drugs, that were excluded from participation during the first year. This adaptation was made due to low inclusion numbers. This was adapted in the manuscript accordingly.
- When you describe patient characteristics. Howe many had frequent hospitalizations or frequent infections?
We collected data on recent infections and hospitalization since we assume this could influence HRQoL. The data were added in the manuscript. The missing data in our sample were possibly related to a recent infection or hospitalization during COVID pandemic. In our sample we could not further analyze this possible influential variable.
- Were some patients with higher risk MDS evaluated for transplant? How did this affect their disease perception?
Two patients were evaluated for allogeneic transplant, one patient receiving intensive chemotherapy, another one receiving Vidaza. We did not collect data on QoL following transplant since these patients present with peculiar treatment-specific side effects that influence HRQoL.
5- In the discussion, instead of referring to the questionnaire items with numbers try to explain what this means so that readers don't have to go back and forth.
This was adapted accordingly.